

# Distribution and ecological segregation on regional and microgeographic scales of the diploid *Centaurea aspera* L., the tetraploid *C. seridis* L., and their triploid hybrids (Compositae)

Alfonso Garmendia[1], Hugo Merle[2], Pablo Ruiz[2] and Maria Ferriol[1]

[1] Instituto Agroforestal Mediterráneo, Universitat Politècnica de València, Valencia, Spain
[2] Departamento de Ecosistemas Agroforestales, Universitat Politècnica de València, Valencia, Spain

## ABSTRACT

Although polyploidy is considered a ubiquitous process in plants, the establishment of new polyploid species may be hindered by ecological competition with parental diploid taxa. In such cases, the adaptive processes that result in the ecological divergence of diploids and polyploids can lead to their co-existence. In contrast, non-adaptive processes can lead to the co-existence of diploids and polyploids or to differentiated distributions, particularly when the minority cytotype disadvantage effect comes into play. Although large-scale studies of cytotype distributions have been widely conducted, the segregation of sympatric cytotypes on fine scales has been poorly studied. We analysed the spatial distribution and ecological requirements of the tetraploid *Centaurea seridis* and the diploid *Centaurea aspera* in east Spain on a large scale, and also microspatially in contact zones where both species hybridise and give rise to sterile triploid hybrids. On the fine scale, the position of each *Centaurea* individual was recorded along with soil parameters, accompanying species cover and plant richness. On the east Spanish coast, a slight latitudinal gradient was found. Tetraploid *C. seridis* individuals were located northerly and diploid *C. aspera* individuals southerly. Tetraploids were found only in the habitats with strong anthropogenic disturbance. In disturbed locations with well-developed semi-fixed or fixed dunes, diploids and tetraploids could co-exist and hybridise. However, on a fine scale, although taxa were spatially segregated in contact zones, they were not ecologically differentiated. This finding suggests the existence of non-adaptive processes that have led to their co-existence. Triploid hybrids were closer to diploid allogamous mothers (*C. aspera*) than to tetraploid autogamous fathers (*C. seridis*). This may result in a better ability to compete for space in the tetraploid minor cytotype, which might facilitate its long-term persistence.

Corresponding author
Maria Ferriol,
mafermo@upvnet.upv.es

## INTRODUCTION

Polyploidy is considered a ubiquitous process that has played a key role in plant evolution (*Wendel et al., 2016*). Whether polyploidy represents an evolutionary advantage or disadvantage remains unclear, and mainly depends on taxa (*Madlung, 2013*) and evolutionary moment (*Parisod, Holderegger & Brochmann, 2010*). Obstacles to new polyploids establishment include ecological and reproductive competition with parental diploid species (*Petit, Bretagnolle & Felber, 1999*). However, the coexistence of diploids and polyploids can be fairly stable for different factors, of which ecological divergence following adaptive processes is among the most important (*Mable, 2003*; *Hülber et al., 2009*; *Mráz et al., 2012*). This divergence is driven by an environmentally-dependent selection along an abiotic or biotic gradient, which results in the differentiation of the ecophysiological requirements of different related cytotypes. Polyploids may display a better fitness advantage in novel environments due to both increased genetic diversity, on which selection can act, and novel biochemical pathways and transgressive characters (*Leitch & Leitch, 2008*). This new steady state may confer onto them a predisposition towards both the extension of their ecological amplitude and quick adaptation to changing environmental conditions (*Ramsey, 2011*; *Hülber et al., 2015*). As a result, polyploids might respond better to aridity (*Manzaneda et al., 2012*; *McAllister et al., 2015*), higher or lower altitudes and latitudes (*Hardy et al., 2000*; *Sonnleitner et al., 2010*; *Zozomová-Lihová et al., 2015*), lower temperatures (*Zozomová-Lihová et al., 2015*; *Paule et al., 2017*), salt (*Chao et al., 2013*), and limiting soil characteristics (*Kolář et al., 2013*). However, this is not always consistent and, in some cases, a wider ecological amplitude has been found in diploids compared to tetraploids (*Španiel et al., 2008*; *Theodoridis et al., 2013*). Differentiation among related cytotypes can be reflected in shifts in the abundance of accompanying species (*Johnson, Husband & Burton, 2003*), spatial segregation based on distinct ecological preferences within the same habitat type (*Raabová, Fischer & Münzbergová, 2008*), or separation of cytotypes into plant communities that differ in structure and physiognomy (*Lumaret et al., 1987*).

The co-existence of individuals of different ploidy levels can also be caused by non-adaptive processes, such as the recent origin of polyploids in primary contact zones (*McArthur & Sanderson, 1999*), multiple polyploidisation events (*Leitch & Bennett, 1997*), and the predominance of vegetative reproduction associated with local dispersal in polyploids (*Šafářová & Duchoslav, 2010*). In contrast, non-adaptive processes may lead to differentiated distributions, even in those cases where cytotypes have similar ecological requirements. The minority cytotype disadvantage (*Levin, 1975*) is a particular concern. This occurs mainly in contact zones where different cytotypes of the same or closely-related species produce hybrid offspring, which are generally triploid individuals that are mostly sterile and act as a major reproductive barrier (*Petit, Bretagnolle & Felber, 1999*; *Husband, 2004*; *Herben, Trávníček & Chrtek, 2016*). By assuming random mating, it can be stated that the lower the frequency of a cytotype, the higher the proportion of its ineffective pollinations. For each generation, the minority cytotype produces proportionally fewer offspring than the majority cytotype, which leads to its progressive
elimination (*Baack, 2004*). Differentiated distributions of related cytotypes can also be achieved through historical colonisations and past dispersals (*Kolář et al., 2009*), and through variations in mating and competition patterns (*Trávníček et al., 2011*).

As a result of these adaptive and non-adaptive processes, distribution of well-established neopolyploids tend to separate from that of their diploid ancestors, although overlapping areas may exist with varying magnitudes depending on taxa (*Thompson, Husband & Maherali, 2015*; *Zozomová-Lihová et al., 2015*). Most studies that deal with distributions and ecological affinities of related cytotypes have been assessed by comparing single-ploidy level populations or by broad-scale surveys of individuals (*Balao et al., 2009*; *Koutecký, Štěpánek & Baďurová, 2012*; *Krejčíková et al., 2013*; *McAllister et al., 2015*). In most of them, individuals of different ploidy levels appear to occupy differentiated geographical and/or ecological areas. In particular, allopolyploids more frequently display an intermediate niche between those of their diploid progenitors, and also a wider niche overlap with them (*Blaine, Soltis & Soltis, 2016*). In contrast, much fewer surveys that deal with the segregation of sympatric populations of different ploidy levels on fine scales have been performed. As most ecological variables are spatially structured, these studies can allow inference of whether one single microhabitat is suited for different related cytotypes (hybrid zones, e.g. *Baack & Stanton, 2005*; *Kolář et al., 2009*), or if differentiated habitats suited for a single cytotype are microspatially segregated (mosaic zones, e.g. *Suda et al., 2004*; *Hülber et al., 2015*). Furthermore, if heteroploid hybridisation is possible, the hybrids that emerge in contact zones have to establish and compete with parental individuals. Their persistence may be influenced by the magnitude of ecological differentiation from parental populations, and by their geographical and ecological position in relation to those of their parents (*Ståhlberg & Hedrén, 2009*).

*Centaurea* (Compositae) is a recent, taxonomically intricate genus due to the existence of polyploidy, descending dysploidy cycles, and hybridisation events (*Hellwig, 2004*; *Romaschenko et al., 2004*). *Centaurea aspera* L. and *Centaurea seridis* L. are perennial herbaceous plants that belong to the section Seridia (Juss.) Czerep. *C. aspera* is widespread from south–west Europe (it extends eastwardly to central Italy) to north–west Africa (*Tutin & Heywood, 1976*; *Devesa, 2016*). It is highly differentiated locally and grows in a wide range of habitats: in dry and open habitats at low elevations, remnant Mediterranean forest patches, and nitrophilous sand dunes. In Europe, only diploid populations of *C. aspera* have been recorded (compiled in *Invernón, Devesa & López, 2013*; see also *Garmendia et al., 2015*). *C. seridis* is an allotetraploid that derives from *C. aspera* and one still unknown closely-related species (*Invernón, Devesa & López, 2013*; *Ferriol, Merle & Garmendia, 2014*). It has a narrower distribution from south–east Spain to north–west Africa (*Tutin & Heywood, 1976*), although it has also been cited as a rare species in Italy (the Calabria region and Sicily, *Conti et al., 2005*), Albania and Greece (*Gibbons, 2003*; *Devesa, 2016*). It usually develops on maritime sand soils and rarely occurs inland, on rocky soils in dry open habitats. In east Spain and west Morocco, both species co-exist in several contact zones, hybridise and generate morphologically intermediate hybrids, *C. x subdecurrens* Pau (*Ferriol et al., 2012*, *Garmendia et al., 2015*). In east Spain, the hybrids from diploid subspecies of *C. aspera* and tetraploid *C. seridis* are triploid and

sterile (*Ferriol, Merle & Garmendia, 2014*). To date in Spain, six contact zones have been described in sand and pebble coastal dunes between north Castellón and Almería (Calblanque, Guardamar del Segura, Santa Pola, El Saler, Marjal dels Moros, and Chilches) (*Garmendia et al., 2010*), and one inland (Sax) (*Merle, Garmendia & Ferriol, 2010*). In all the 165 individuals previously evaluated in these six contact zones, ploidy level determined by flow cytometry unambiguously corresponded to the morphological characters that are discriminant of each taxon (*C. aspera* $2x = 22$, *C. seridis* $4x = 44$, and *C. x subdecurrens* $3x = 33$) (*Ferriol et al., 2012*; *Ferriol, Merle & Garmendia, 2014*). No ploidy levels higher than tetraploid were found.

Both *C. aspera* and *C. seridis* are insect-pollinated and their flowering periods overlap widely in east Spain (*Bosch, Retana & Cerdá, 1997*; *Ferriol et al., 2015*). However, while diploids are strictly allogamous and do not display mentor effects, tetraploids are highly autogamous (*Ferriol et al., 2015*). Consequently, hybrids asymmetrically form: all triploid intact cypselae come from the diploid mothers pollinated by the pollen of tetraploids. In artificial crosses between *C. aspera* and *C. seridis*, only triploids were observed in the progeny (*Ferriol et al., 2015*). No tetraploids, which could act as interploid bridge, were found to form from unreduced gametes from the diploid *C. aspera* (*Sutherland & Galloway, 2017*). These unidirectional crossings could help *C. seridis* overcome the minority cytotype exclusion effect to enhance its short-term survival (*Van de Peer, Mizrachi & Marchal, 2017*).

In this study, we analysed the spatial distribution of diploid *C. aspera*, triploid *C. x subdecurrens*, and tetraploid *C. seridis*, and we tested the hypothesis that they are ecologically differentiated, both on a broad scale and microspatially in contact zones where they grow in sympatry. This potential geographic and/or ecological segregation may contribute to interspecific reproductive isolation. Specifically, we addressed the following questions: (i) what is the spatial structure and what are the ecological requirements of diploids and their allotetraploid derivatives across east Spain? (ii) What is the microspatial distribution pattern of individuals in mixed-ploidy plots? (iii) Does the distribution of triploids in contact areas correspond to patterns of crossability between the diploid *C. aspera* and the tetraploid *C. seridis*? (iv) Is there any correlation between taxa distribution on a fine scale and ecological microhabitat characteristics? In coastal dunes, there are strong gradients of various environmental factors that run perpendicularly to the shoreline. These include sand grain diameter, wind-driven sand movement, amount of salty spray, water availability, nutrient level, soil pH, vegetation cover, and plant diversity (*Brown & McLachlan, 1990*; *Brunbjerg et al., 2012*). As these factors are expected to act as filtering processes, we predicted that, if taxa are ecologically differentiated, their habitat should differ from them. Altogether, these questions can shed light onto long-term diploids/polyploids co-existence, and whether it is a result of adaptive *vs.* non-adaptive mechanisms.

## MATERIALS AND METHODS

### Population sampling and ecology on a broad scale

*Centaurea* individuals were sampled on two geographic scales. On a broad scale, extensive sampling was conducted in east and south Spain, and in south France, during the

2008–2015 period. Here, we focused particularly on the Mediterranean coast as it is the typical habitat of *C. seridis*, where most contact zones with *C. aspera* occur with formation of triploid hybrids. A total of 39 sites were selected from the ANTHOS project and the BDNFF botanic databases (*ANTHOS, 2018*; Base de Données nomenclaturales de la Flore de France (*BDNFF, 2018*)). In each location, several environmental parameters were recorded: coastal urbanisation (presence of buildings, roads, promenades), anthropic disturbance (human traffic, tourism, grazing), presence of large salt marshes near the sea that prevent the presence of well-developed semi-fixed and fixed dunes, soil type (sand dune, fossil dune, pebble dune), and vegetation type.

## Population sampling on the microspatial scale

Three coastal contact zones, where *C. aspera*, *C. seridis*, and *C. x subdecurrens* were present, were selected to assess microspatial distribution. These contact zones corresponded to sites 14 (Marjal dels Moros), 15 (El Saler North), and 16 (El Saler South), the first on pebble dunes and the last two on sandy soils (Table 1). The limits of each sampling plot were determined by geo-referencing corners and using ropes between them (Table 2). Each plot was of an appropriate size to include more than 50 individuals of each parental taxon (*C. aspera* 2x and *C. seridis* 4x). Samplings were performed in spring (March and April 2013). The exact location of each individual in each plot was determined by a Garmin Etrex GPS(Olathe, KS, USA). All the locations from each plot were collected with the same GPS receiver in the shortest possible sampling time (within 3 days).

## Ecological differentiation of taxa on the microspatial scale

To compare the ecological requirements of *C. seridis*, *C. aspera*, and *C. x subdecurrens*, we selected the "El Saler North" plot because of its regular shape, high individual density, and the absence of strong discontinuities due to pathways or other infrastructures. In the field, a grid (20 × 120 m) was laid out with an E–W orientation and perpendicular to the shoreline, which was subdivided in 2,400 quadrats of 1 $m^2$ delimited by ropes.

In the central quadrat of each 25 $m^2$ area (5 × 5 m) (96 quadrats in all), the following parameters were measured: (1) total vegetation cover, specifically the cover of chamaephytes, hemicryptophytes, geophytes, and therophytes, (2) distance to the nearest pathway (or percentage of quadrat occupied by the pathway), (3) slope aspect, (4) slope inclination, (5) plant species richness, (6) occurrence (presence/absence) of species present in the plot.

Soil parameters were analysed in the centre of each 100 $m^2$ area, which was already delimited by ropes (24 samples in all). Soil was collected manually with a soil core sampler (15 cm deep and 10 cm in diameter). Soil samples were air-dried at 25 °C and passed through a 2 mm sieve. Grain sizes were determined by dry sieving, using five sieve intervals from 2 to 0.05 mms. Soil pH was determined with a soil-distilled water ratio of 1:2.5 w/v. Soil organic matter was determined by potassium dichromate oxidation (*Nelson & Sommers, 1996*). Electrical Conductivity (EC) was measured by an EC meter on 50 ml of a 1:5 w/v soil to water extract, to which two drops of 0.1% sodium hexametaphosphate were added (*MAPA, 1986*). Soil samples were also defined for their

**Table 1 Single-ploidy populations (*C. aspera 2n* and *C. seridis 4n*) and mixed-ploidy populations (*C. aspera*, *C. seridis*, and *C. x subdecurrens 3n*) in easter and southern Spain and in southern France.**

| N | Geographic coordinates | Site | Urbanisation at <500 m from the sea | Anthropic disturbance | Salt marshes | Dune type/Inner land | Vegetation type and habitat. | C. aspera abundance | C. x subdecurrens abundance | C. seridis abundance |
|---|---|---|---|---|---|---|---|---|---|---|
| 1 | 43°27'30.1788"N 4°28'16.9212"E | Camargue | No | Low | No | Sand | *Centaureo maritimae–Echietum sabulicolae* | Common | – | – |
| 2 | 43°14'12.0588"N 6°39'46.5012"E | Pampelone | Yes | High (T) | No | Sand | *Centaureo maritimae–Echietum sabulicolae* | Common | – | – |
| 3 | 43°9'15.2250"N 2°57'48.0300"E | Narbonne | Yes | High (R) | No | Inner land | Open shrubland near the road and parking area. | Common | – | – |
| 4 | 43°7'30.4212"N 6°21'47.1600"E | Lavandou | Yes | High (T) | No | Sand | *Centaureo maritimae–Echietum sabulicolae* | Common | – | – |
| 5 | 43°6'27.1800"N 6°10'49.7388"E | Hyeres | Yes | High (T) | No | Sand | *Centaureo maritimae–Echietum sabulicolae* | Common | – | – |
| 6 | 42°15'11.8800"N 3°8'26.0100"E | Roses | No | High (T) | No | Sand | *Centaureo maritimae–Echietum sabulicolae* | Common | – | – |
| 7 | 41°34'21.1188"N 2°25'12.7812"E | Turó d'en Cabanyes | No | Low (R) | No | Inner land | Open shrubland near the road and in the ditch. | Common | – | – |
| 8 | 41°15'59.4200"N 0°53'17.2800"E | Montsant | No | Low | No | Inner land | *Rosmarino-Ericion* and nitrophilous herbs near the road. | Common | – | – |
| 9 | 41°4'2.3988"N 1°4'49.5588"E | Salou | Yes | High (T) | No | Inner land | Open and nitrophilous shrubland, in a disturbed ravine with exotic plants (*Acacia* spp.) | Common | – | – |
| 10 | 41°4'1.2900"N 1°4'50.7450"E | Cambrils | Yes | High (T) | No | Sand | Open and nitrophilous shrubland | Common | – | – |
| 11 | 39°59'29.0688"N 0°1'37.7832"E | Castellón North | Yes | High (T, P) | No | Sand | *Centaureo maritimae–Echietum sabulicolae* | – | – | Common |
| 12 | 39°46'24.4705"N 0°93.2886"W | Chilches | Yes | High (T, G) | No | Pebble | Open and nitrophilous shrubland, with *Glaucium flavum*. | Common | Common | Common |
| 13 | 39°40'45.5988"N 0°12'11.3004"W | Canet de Berenguer | Yes | High (T) | No | Sand | *Centaureo maritimae–Echietum sabulicolae* | – | – | Common |
| 14 | 39°37'23.6111"N 0°15'10.4670"W | Marjal dels Moros | No | High (G) | Yes (150 m from the sea) | Pebble | Open and nitrophilous shrubland with *Glaucium flavum, Halimione portulacoides*, and nitrophilous herbs. | Rare | Common | Common |
| 15 | 39°22'13.5527"N 0°19'16.0197"W | Saler North | Yes | High (T) | No | Sand | *Centaureo maritimae–Echietum sabulicolae* with *Phyllirea angustifolia* and *Pistacia lentiscus*. | Common | Common | Common |

| N | Geographic coordinates | Site | Urbanisation at <500 m from the sea | Anthropic disturbance | Salt marshes | Dune type/Inner land | Vegetation type and habitat | C. aspera abundance | C. x subdecurrens abundance | C. seridis abundance |
|---|---|---|---|---|---|---|---|---|---|---|
| 16 | 0°18′9.7339″W 39°19′38.3832″N | Saler South | No | High (T) | No | Sand | Centaureo maritimae–Echietum sabulicolae with Phyllirea angustifolia and Pistacia lentiscus. | Common | Common | Common |
| 17 | 0°38′36.8400″W 38°56′41.8500″N | Montesa | No | High (R) | No | Inner land | Rosmarino-Ericion and ruderal herbs (Foeniculum vulgare...) near the road. | Common | – | – |
| 18 | 0°52′50.4012″W 38°48′42.1200″N | Font de la Figuera 1 | No | Low | No | Inner land | Ruderal vegetation inside an industrial estate. | Common | – | – |
| 19 | 0°53′9.5388″W 38°47′46.6188″N | Font de la Figuera 2 | No | Low | No | Inner land | Open shrubland within a disturbed ravine in an urbanized area. | – | – | Common |
| 20 | 0°51′38.5812″W 38°39′27.0000″N | Villena 1 | No | Low | No | Inner land | Open and arid shrubland along the road ditch. | Common | – | – |
| 21 | 0°51′2.1600″W 38°34′23.8800″N | Villena 2 | No | Low | No | Inner land | Ruderal vegetation, in an area with debris. | – | – | Common |
| 22 | 0°48′57.4125″W 38°32′28.0375″N | Sax | Yes | High | No | Inner land | Rosmarino-Ericion and nitrophilous herbs near the road, with Pinus halepensis. | Common | Rare | Common |
| 23 | 0°47′13.5000″W 38°29′44.4600″N | Elda | No | Low | No | Inner land | Rosmarino-Ericion and arid shrublands | Rare | Rare | Common |
| 24 | 0°30′56.6750″W 38°13′57.3300″N | Santa Pola | No | High (T) | No | Fossil | Centaureo maritimae–Echietum sabulicolae with some halophytes. | Common | Common | Common |
| 25 | 0°38′33.8250″W 38°7′36.0437″N | Guardamar | No | High (T) | No | Sand | Pinus pinea with open understory | Common | Common | Common |
| 26 | 0°39′21.6000″W 38°0′51.4188″N | La Mata | Yes | High (T, P) | No | Sand | Pinus pinea with open understory and open macchia. | Common | Common | Common |
| 27 | 0°45′25.6212″W 37°48′42.4800″N | San Pedro del Pinatar | No | Low | Yes (at 500 m or less) | Sand | Mobile dunes and salt marsh vegetation | – | – | Common |
| 28 | 0°45′26.7984″W 37°36′35.6004″N | Calblanque | No | Low | No | Fossil | Mayteno-Periplocetum angustifoliae | Rare | Rare | Rare |
| 29 | 1°59′43.7388″W 37°11′57.0012″N | Bédar | No | Low | No | Inner land | Open shrubland near the road, with Pistacia lentiscus and Anthyllis cytisoides. | Common | – | – |
| 30 | 2°24′37.3050″W 36°59′36.5700″N | Alhamilla | No | Low | No | Inner land | Open shrubland along the road ditch | Common | – | – |
| 31 | 3°28′58.3464″W 36°51′58.5216″N | Guadalfeo 1 | No | Low (R) | No | Inner land | Open shrubland (Rosmarino-Ericion) along the road ditch | Common | – | – |

(Continued)

| N | Geographic coordinates | | Site | Urbanisation at <500 m from the sea | Anthropic disturbance | Salt marshes | Dune type/Inner land | Vegetation type and habitat. | C. aspera abundance | C. x subdecurrens abundance | C. seridis abundance |
|---|---|---|---|---|---|---|---|---|---|---|---|
| 32 | 3°28′25.0350″W | 36°51′52.5150″N | Guadalfeo 2 | No | Low (R) | No | Inner land | Open shrubland with *Pinus halepensis* | Common | | |
| 33 | 3°28′25.4388″W | 36°51′51.7788″N | Orgiva | No | Low (R) | No | Inner land | Open shrubland with *Pinus halepensis* near the road and in the road ditch. | Common | – | – |
| 34 | 3°3′46.8612″W | 36°47′2.1588″N | La Parra | Yes | High (R) | No | Inner land | Open shrubland along a forest trail, with ruderal species | Common | – | – |
| 35 | 3°27′38.4012″W | 36°41′48.3612″N | Motril | Yes | High (T) | No | Sand | *Centaureo maritimae–Echietum sabulicolae* | | | Common |
| 36 | 4°51′38.2800″W | 36°30′27.4350″N | Marbella | Yes | High (T) | No | Sand | *Centaureo maritimae–Echietum sabulicolae* | – | – | Common |
| 37 | 5°22′54.2100″W | 36°10′42.7836″N | Puente Mayorga | Yes | High (T) | No | Sand | Small patch of *Centaureo maritimae–Echietum sabulicolae* | Common | – | Rare |
| 38 | 5°20′20.1012″W | 36°9′38.9988″N | Gibraltar | Yes | High (T) | No | Sand | *Centaureo maritimae–Echietum sabulicolae* | – | – | Rare |
| 39 | 5°26′41.1612″W | 36°5′42.7812″N | Algeciras | Yes | High (T) | No | Sand | *Centaureo maritimae–Echietum sabulicolae* | – | – | Common |

**Notes:**
Geographical coordinates and environmental parameters are described for each site.
N, number of site; T, Tourism; R, Road; P, Path; G, Grazing.

 

**Table 2 Geographical location, number of *Centaurea* spp. individuals and ratio between number of individuals of each cytotype in the three sampling plots used for analysing microspatial distribution.**

| Plot | Corners coordinates UTM (WG S84, 30S) | Area (m²) | *Centaurea* individuals | | *C. aspera* (2x) | | *C. seridis* (4x) | | *C. x subdecurrens* (3x) | | Ratio 4x/2x number of individuals | Ratio 3x/2x number of individuals | Ratio 3x/4x number of individuals |
|---|---|---|---|---|---|---|---|---|---|---|---|---|---|
| | | | Number | i/ha | Number | i/ha | Number | i/ha | Number | i/ha | | | |
| Marjal dels Moros | 735806 4389584 735782 4389591 735724 4389480 735749 4389463 735861 4389546 | 7,943 | 501 | 630.7 | 51 | 64.2 | 441 | 555.2 | 9 | 11.3 | 8.64 | 0.18 | 0.02 |
| El Saler North | 730716 4361306 730721 4361286 730858 4361314 730842 4361336 | 3,066 | 352 | 1148.1 | 235 | 766.5 | 81 | 264.2 | 36 | 117.4 | 0.34 | 0.15 | 0.44 |
| El Saler South | 732347 4356538 732383 4356487 732574 4356551 732556 4356609 | 12,826 | 137 | 106.8 | 43 | 33.5 | 82 | 63.9 | 12 | 9.4 | 1.91 | 0.28 | 0.15 |

**Note:**
i/ha, individuals per hectare.

colour indices (Hue, Value, and Chroma) according to the Munsell colour chart under similar illumination conditions, and following *Post et al. (2000)*. These indices have been significantly related with soil parameters, specifically the Munsell Value component with albedo ($R^2$ = 93%, *Post et al., 2000*). Furthermore, digital photographs of each soil sample arranged on a Petri dish were taken in the laboratory with a high quality digital camera (16.1 megapixels) under standard lighting conditions and with no flash, from a height of 12 cm above the sample (resolution 4,608 × 3,456 pixels). Digital images were processed using GIMP 2.8.4 (*GIMP team, 2014*). After calibrating the RGB values following *Levin, Ben-Dor & Singer (2005)*, a region of interest (ROI) covering the central part of the Petri dish with the soil sample was defined and the RGB coordinates of the ROI were obtained. Redness Index [RI = $R^2/(B^*G^3)$], which correlates highly with the free iron content in sand dune soils ($R^2$ = 88.9%, *Levin, Ben-Dor & Singer, 2005*), was calculated.

In all the studied quadrats, the presence or absence of all three taxa was recorded to describe and compare their representative microhabitats.

## Statistical analyses

To render this document, we used R, R Markdown, Knitr, and Pandoc (*R Core Team, 2017*; *Xie, Hill & Thomas, 2017*). We also used packages readxl (*Wickham & Bryan, 2017*) and writexl (*Ooms, 2017*) to import and export data, and dplyr (*Wickham et al., 2017*) and tidyr (*Wickham & Henry, 2018*) to manage data.

*Centaurea* individuals were plotted on maps on the microspatial scale using maptools (*Bivand & Lewin-Koh, 2017*) and rgdal (*Bivand et al., 2017*). The package ggsn (*Baquero, 2017*) was also employed to include scales on some maps.

In the mixed-ploidy plots, spatial distribution and the relationship between species were analysed following *Pebesma & Bivand (2005)* using spatstat (*Baddeley & Turner, 2005*; *Baddeley, Rubak & Turner, 2015*). The distribution pattern of the individuals within each taxon was analysed by Ripley's *K*-function (*Ripley, 1976*). The *K*-function determines the distribution pattern (clumped, random, or regular) by counting the number of conspecific individuals within a given radius *r* of each individual in the study area, and by comparing the mean number with the counts that derived from the density of this species in the plot. The results were compared with those observed with random Poisson distribution confidence intervals, which were obtained by a Monte Carlo test with 300 independent repeats per plot. Therefore, when the observed *K*(*r*) was over the confidence interval, distribution was considered clumped. If it was under the confidence interval, it was considered regular. When it was between the limits of the interval, it was not separated from random.

Pairwise interspecific associations were examined by Chi-squared tests following *Baddeley (2010)*. By considering two taxa, each sampling plot was divided into three equal-area density (low, intermediate and high) levels for the first taxon, and the frequencies (quadrat countings) of the second taxon in each equal-area were established. A Chi-squared test was performed to determine the significance of the individual distribution of the second taxon along the different densities areas of the first taxon.

Comparisons were made both between parentals (*C. aspera* 2*x* and *C. seridis* 4*x*), and also between the hybrid (*C. x subdecurrens* 3*x*) and each parental.

The ecological differentiation among quadrats was summarised by multivariate techniques in vegan for R (*Oksanen et al., 2009*). Initially, a non-metric multidimensional scaling (NMDS) analysis was run to examine the distribution of the species composition in the different quadrats using 200 random starts. The multidimensional space of species accompanying the *Centaurea* taxa, represented by pairwise Bray–Curtis distances between individuals, was reduced to a four-dimensional configuration (NMDS-space), and the quality of this transformation was indicated by a non-linear monotone transformation of the observed distances and ordination distances called "stress" (*Oksanen, 2009*). The ordination result was post-processed with the "metaMDS" (default-options) function, which repeats calculations 20 times with random starting arrangements (*Oksanen et al., 2009*). The configuration with the lowest stress for the given number of axes was chosen. The results were scaled to make interpretation easier, and the ecological variables were fitted over the first two axes. To analyse the differentiation among quadrats according to soil parameters, an NMDS was performed similarly using the results of the soil variables instead of species. Additionally, for each *Centaurea* taxon, the ecological characteristics between quadrats with and without plants of the analysed taxon were compared by non-parametric analyses (Wilcoxon signed rank test for continuous variables and Chi-square for categorical variables), followed by Bonferroni correction. A statistically significant difference was considered if $P \leq 0.05$.

The packages ggmap and ggplot2 (*Wickham, 2016*; *Kahle & Wickhman, 2013*) were used to plot graphics.

## RESULTS

### Distribution of the *Centaurea* taxa in east Spain and south France, and ecological preferences

The occurrence of the diploid *C. aspera* and/or tetraploid *C. seridis* individuals in all the 39 studied sites is shown in Fig. 1 and Table 1. The two taxa showed contrasting distribution patterns on a broad scale.

*Centaurea seridis* is present within a range that goes from the surroundings of Castellón, which represents its north limit, to the Strait of Gibraltar, which separates the Mediterranean Sea and the Atlantic Ocean where it is absent (Fig. 1). This species is a coastal dune specialist. Plants grow on mobile to fixed dunes, with both sandy and stony soils, regardless of the presence of inter- and post-dune salt marshes. They are especially abundant in disturbed dunes due to human traffic and grazing that result in removed soils, absence of a dense vegetation cover, and presence of nitrophilous species. Rarely, in only four locations, whose distance from one another is <38 km (Font de la Figuera, Villena, Sax, and Elda), *C. seridis* was also found inwardly in disturbed shrublands, along with ruderal and nitrophilous species.

*C. aspera* is mainly an inland species with a broader distribution area (Fig. 1). On the Mediterranean coast, *C. aspera* individuals were found on the semi-fixed and fixed dunes

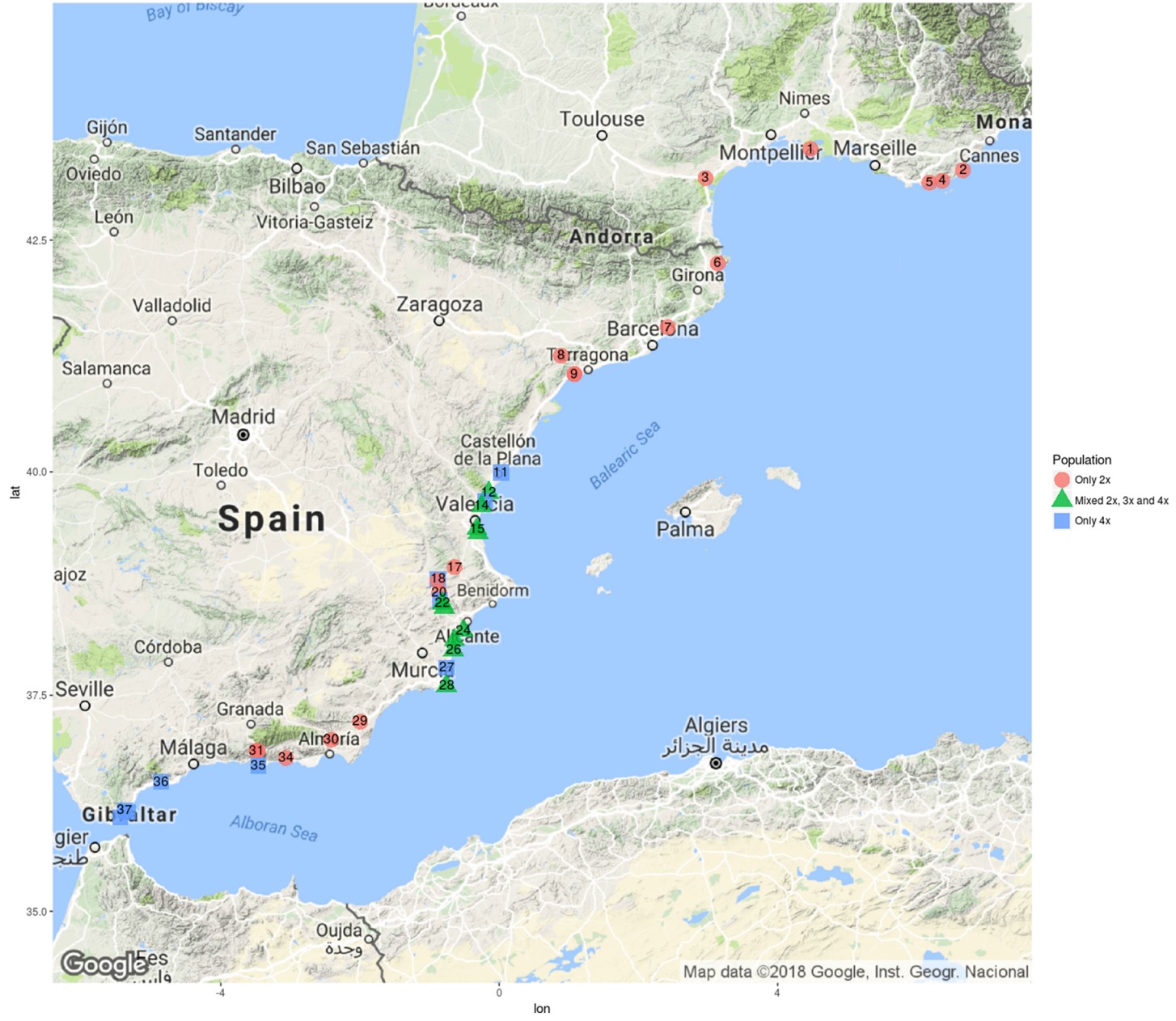

**Figure 1 Localization of single-ploidy populations and mixed-ploidy populations in east and south Spain and in south France.** Populations of diploid *C. aspera* are represented in red circles, populations of tetraploid *C. seridis* in blue squares, and mixed populations of *C. aspera*, *C. seridis*, and triploid *C. x subdecurrens* in green triangles. Numbers correspond to sites from Table 1. Map by Map Data ©2018 Google, Instituto Geográfico Nacional.

from France to Murcia. However, they were absent at the sites where urbanisation and/or the presence of large salt marshes prevented the occurrence of well-developed semi-fixed and fixed dunes. In Andalusia, *C. aspera* was not found on the Mediterranean coast, but was present in nearby low mountains, with a ruderal and nitrophilous character.

Contact zones between diploids and tetraploids occur from Chilches to Calblanque with the presence of triploids. We found eight contact zones in coastal habitats and

two inland. In the coastal contact zones, *C. aspera* and *C. seridis* co-exist in dune habitats that include well-developed semi-fixed or fixed dunes with the presence of open shrublands and pine forests. The habitat at all sites was disturbed by beach tourists or grazing. As a result, *C. seridis* is frequently found along the abundant pathways that move inland, while *C. aspera* also moves in a seaward direction along the same pathways, and both act as ruderal species. In these situations, the triploid hybrids of *C. x subdecurrens* arise. Similarly, in inland contact zones, diploids and tetraploids co-exist in ruderal plant communities near roads, which causes *C. x subdecurrens* to appear.

## Distribution on the microspatial scale

The three mixed ploidy-plots investigated on the microspatial scale differed in plant abundances, densities, and in proportions of diploids, tetraploids and triploids (Table 2; Fig. 2). In agreement with the biogeography on a broader scale, a lower density of *Centaurea* individuals was found in "El Saler South," which represented the dune habitat with the least anthropic disturbance. Both "Marjal dels Moros," with a reduced area of semi-fixed and fixed pebble dunes, and "El Saler South," with the least anthropic disturbance, displayed a low and similar density of *C. aspera* and *C. x subdecurrens*, while "El Saler North," with strong anthropic disturbance and well-developed semi-fixed dunes, showed a higher density of both diploids and triploids. *C. seridis* displayed the highest density at the most disturbed site ("Marjal dels Moros"), and the lowest at the least disturbed site ("El Saler South"). These results suggest that the extent of semi-fixed dunes and of anthropic disturbance is a more determining factor on the presence of *Centaurea* individuals than dune soil type (pebble *vs.* sand). Consequently, and according to these two factors, the ratio between tetraploids and diploids vastly varied among sites. "Marjal dels Moros" (narrow area of semi-fixed dunes and high disturbance) showed the highest $4x/2x$ ratio, which is considerably higher than "El Saler North" and "El Saler South." The number of triploids was related more to the number of diploids than to the number of tetraploids across sites. Accordingly, the $3x/2x$ ratio was more constant than the $3x/4x$ ratio.

Ripley's $K$-function revealed a clumped distribution of diploids and tetraploids at all the sites (Fig. 3). Triploids also displayed clumped distribution at sites "El Saler North" and "El Saler South," but not at "Marjal dels Moros," where distribution was random. However, "Marjal dels Moros" was the site that included the fewest *C. x subdecurrens* individuals (9). Therefore, this result should be interpreted with caution.

More profound insight into the spatial structure was provided by the Chi-squared tests. When each plot was divided into three equal areas of high, intermediate and low densities of *C. aspera, C. seridis* showed a different distribution pattern depending on the studied plot (Fig. 4). At "El Saler South," no significant differences in the number of tetraploid *C. seridis* individuals were found among several *C. aspera* ($2x$) densities. Furthermore, at "Marjal dels Moros," *C. seridis* individuals were significantly more abundant in the areas with intermediate and high *C. aspera* densities, and were more abundant at "El Saler South" in high and low *C. aspera* densities. A similar variable pattern was observed in *C. aspera* distribution over different *C. seridis* densities.

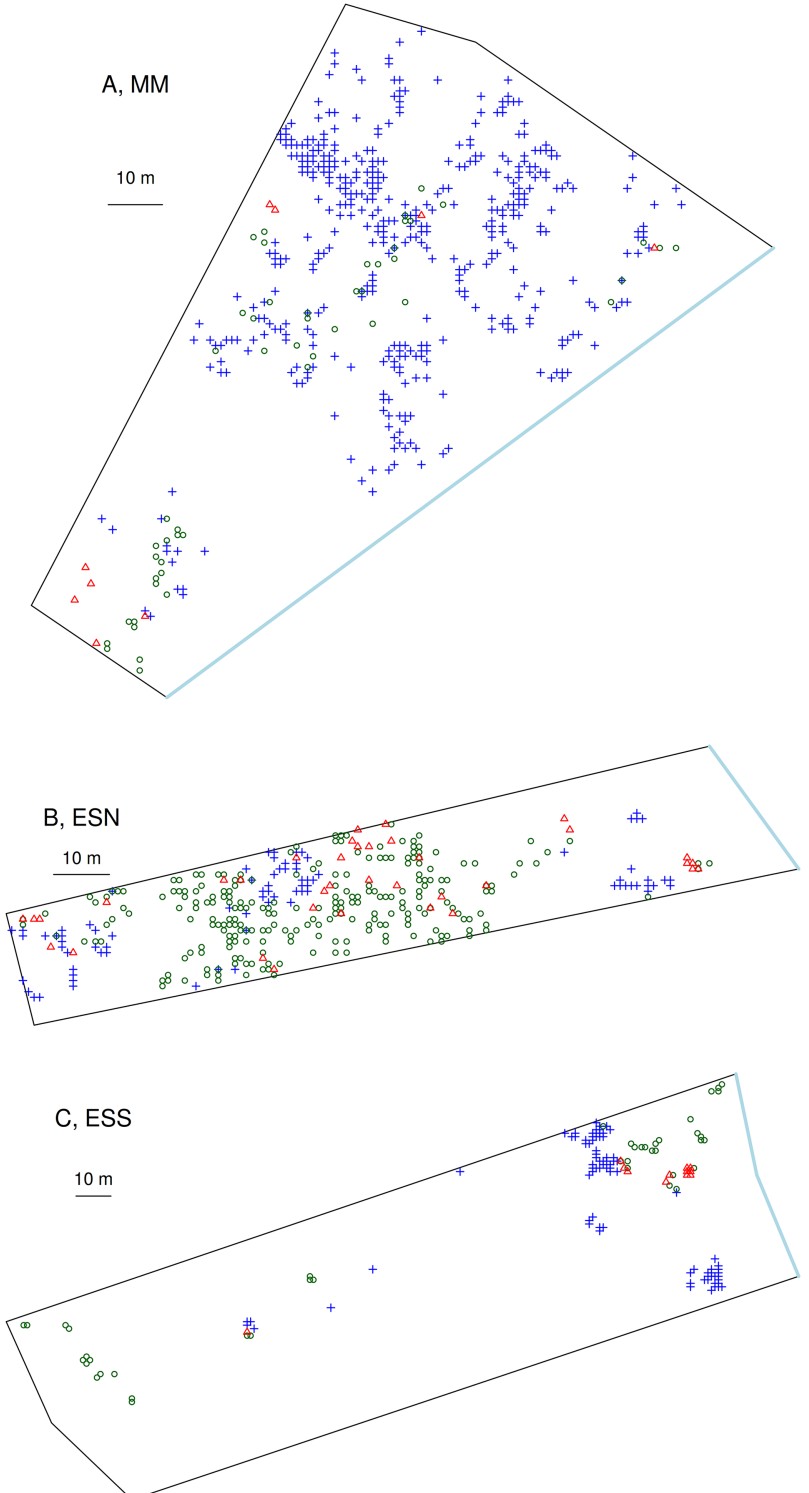

**Figure 2 Fine scale distribution of *C. aspera* (2x), *C. seridis* (4x), and *C. x subdecurrens* (3x) in the three studied sampling plots.** *C. aspera* (2x): green circles, *C. seridis* (4x): blue crosses, and *C. x subdecurrens* (3x): red triangles. Study sites: (A) MM, Marjal dels Moros; (B) ESN, El Saler North; (C) ESS, El Saler South. The blue line represents the edge of the sea.

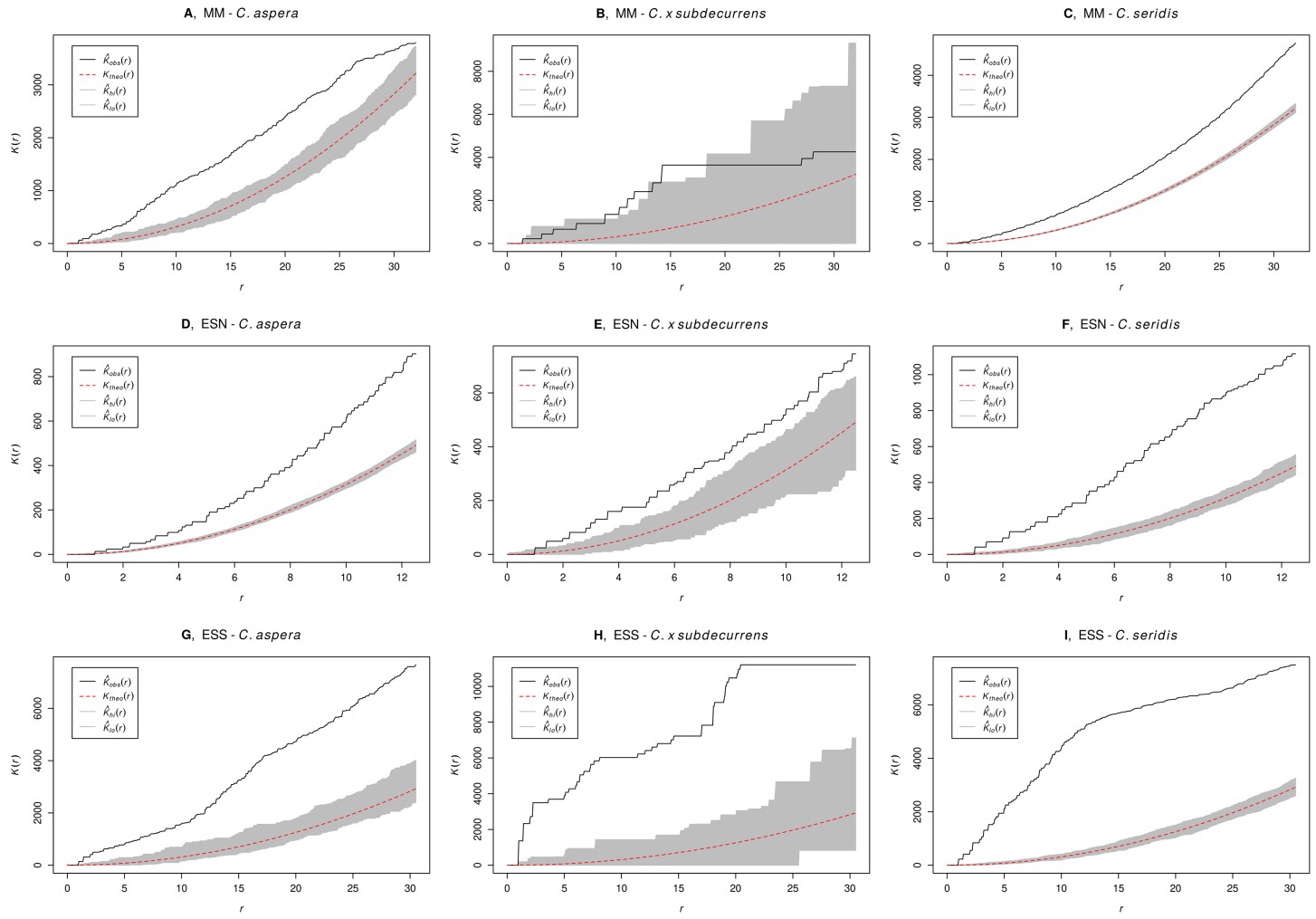

**Figure 3** **Ripley's *K* observed and expected with random distribution for the combined data of all plots, with confidence intervals calculated using Monte Carlo simulations for a Poisson distribution.** Ripley's *K* observed: black, Ripley's *K* expected with random distribution: red, confidence intervals: grey. Values larger than the upper confidence limit indicate significant intracytotype aggregation at the particular distance of *r*. (A–C) Marjal dels Moros (MM), (D–F) El Saler North (ESN), and (G–I) El Saler South (ESS). (A, D, G) *C. aspera*, (B, E, H) *C. x subdecurrens*, and (C, F, I) *C. seridis*.

At "El Saler North," *C. aspera* individuals were significantly more abundant in intermediate and low *C. seridis* densities, while the opposite occurred at "El Saler South," where they were more abundant in areas with a high density of *C. seridis*. Non-significant differences were observed at "Marjal dels Moros." Therefore, no consistent *C. aspera* distribution pattern was found according to that of *C. seridis*, and vice versa. As expected, triploid hybrids were generally more abundant in those areas with high densities for both parentals (Fig. 4). However, these differences were more significant when triploid *C. x subdecurrens* abundance was compared over the several densities of *C. aspera* than those of *C. seridis*. At "Marjal dels Moros," the difference in the number of hybrids among the areas of varying *C. seridis* densities was non-significant.

This higher affinity between *C. x subdecurrens* triploids and *C. aspera* diploids than between triploids and *C. seridis* tetraploids was also supported by the spatial correlograms

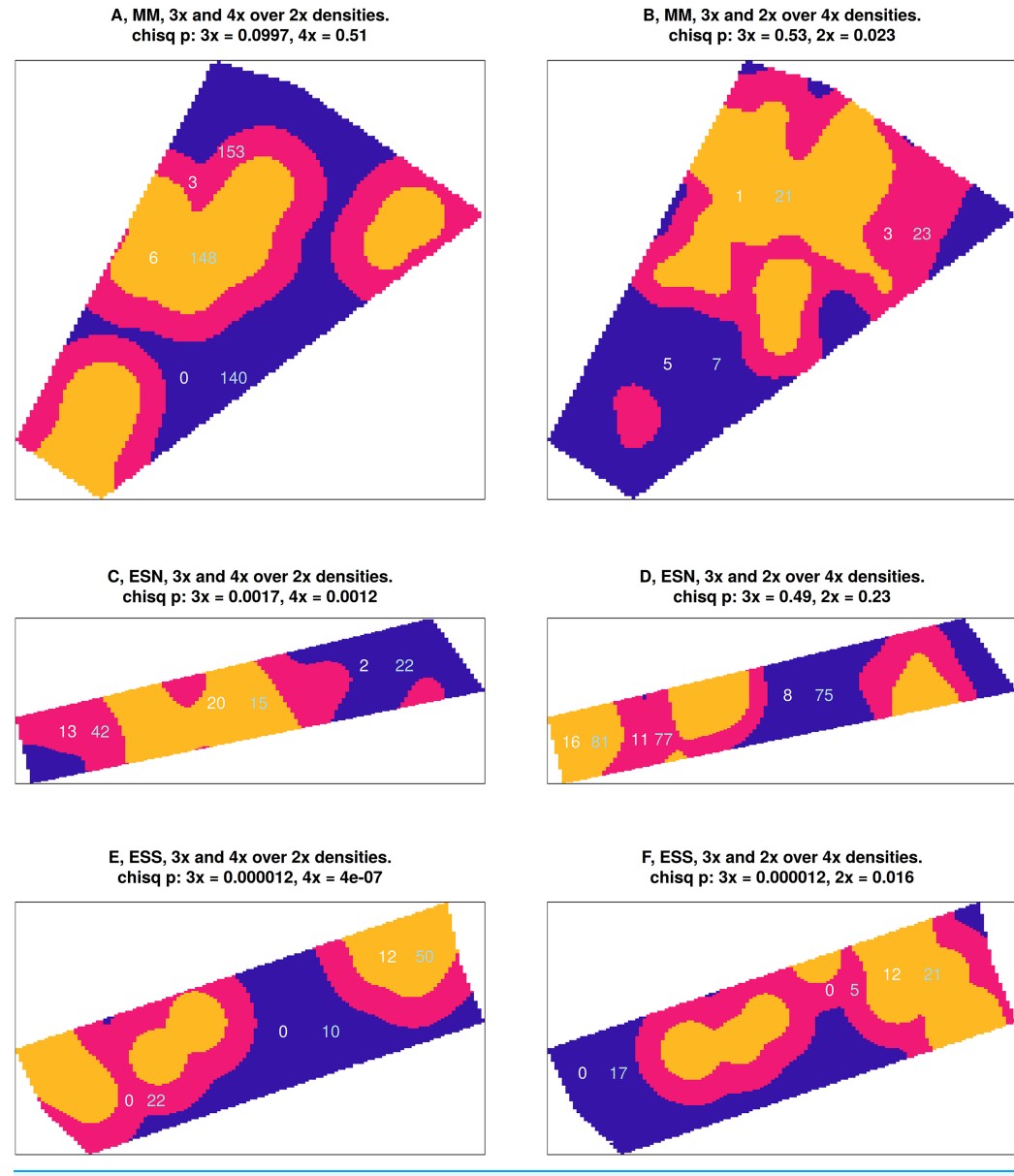

**Figure 4** **Relationship between the abundance of *C. x subdecurrens* (3x) and both parentals (numbers in white) and *C. aspera* (2x) and *C. seridis* (4x) (numbers in light blue) at three sampling plots.** (A and B) Marjal dels Moros (MM); (C and D) El Saler North (ESN), and (E and F) El Saler South (ESS). (A, C, E) The number of *C. x subdecurrens* and *C. seridis* individuals over different densities of *C. aspera*, and (B, D, F) the number of *C. x subdecurrens* and *C. aspera* individuals over different densities of *C. seridis*. Significant Chi-square differences between number of individuals of one taxon over the three equal-area density parts (yellow: high, pink: medium, and blue: low density of the other taxon) are marked.

(Fig. 5). At both "El Saler North" and "El Saler South," the highest *C. x subdecurrens* density came closer to *C. aspera* individuals (at 1.32 ± 0.16 m and 1.43 ± 0.22 m, respectively) than to *C. seridis* individuals (at 7.72 ± 1.04 m and 4.97 ± 0.70 m, respectively). At "Marjal dels Moros," the highest *C. x subdecurrens* density was observed at 5.53 ± 1.43 m from *C. aspera* individuals, whereas two triploid density peaks related to

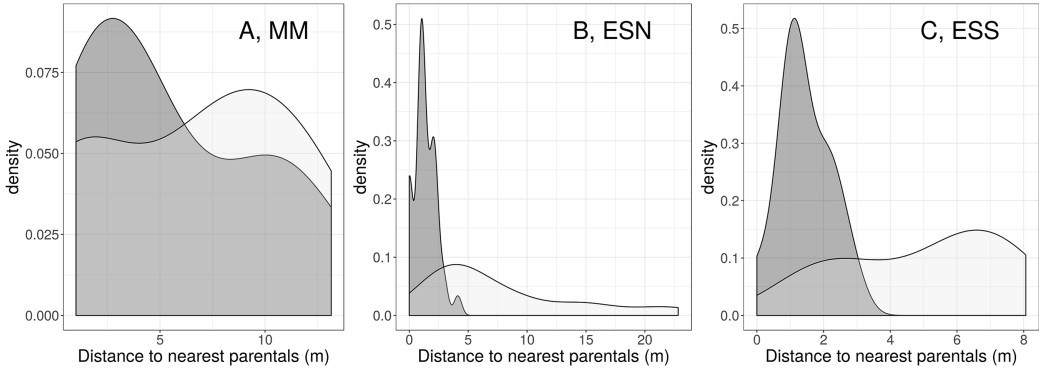

**Figure 5** *C. x subdecurrens* **density (number of individuals per m$^2$) related to the distance between each *C. x subdecurrens* individual and the nearest individual of *C. aspera* and *C. seridis.*** *C. aspera:* dark grey, *C. seridis:* light grey. Study sites: (A) MM, Marjal dels Moros; (B) ESN, El Saler North; (C) EES, El Saler South.

the distance to *C. seridis* individuals were found at 1.22 and 9.92 m, with a mean distance of 6.60 ± 0.57 m. However, the fact that "Marjal dels Moros" displayed the highest tetraploid:diploid individuals ratio, and that *C. seridis* tetraploids were more abundant when *C. aspera* diploids displayed intermediate and high densities, can explain the presence of these two peaks. The correlograms also showed that the difference between the distance from a *C. x subdecurrens* individual to the nearest *C. aspera* individual, and that to the nearest *C. seridis* individual, was bigger at the least disturbed site ("El Saler South") than at the most disturbed sites.

## Ecological differentiation on the microspatial scale

The results of the NMDS performed to analyse the vegetation differentiation among quadrats with the presence/absence of each taxon are shown in Fig. 6. As a whole, plant species composition of the sampled quadrats at "El Saler North" was most variable. Some factors to explain this variability include total vegetation cover, chamaephyte cover, therophyte cover, geophyte cover, presence of trails, and species richness (see Figs. S1–S3 to view the vegetation structure and paths). They all had a relatively strong impact on the differentiation patterns, shown by the length of the vectors in Fig. 6. Nevertheless, based on these vegetation data and environmental variables, it was not possible to ecologically differentiate the *C. aspera* and *C. seridis* individuals as they appeared to be highly intermingled, and showed no clear distribution pattern. Nor was it possible to differentiate *C. aspera* and *C. seridis* individuals according to soil variables, appearing intermingled in the NMDS analysis (Fig. 7).

These results are supported by the pairwise comparisons made between quadrats with the presence and absence of each taxon, and by considering environmental, species, and soil variables. Only some environmental variables significantly differed between quadrats with the presence *vs* absence of *C. aspera* individuals, but none of these significant differences were conclusive (Table 3). Species richness was lower in the quadrats where diploid individuals were absent than in those where they were present (Table 3). Similarly, the hemicryptophyte cover percentage was also significantly lower in the quadrats where

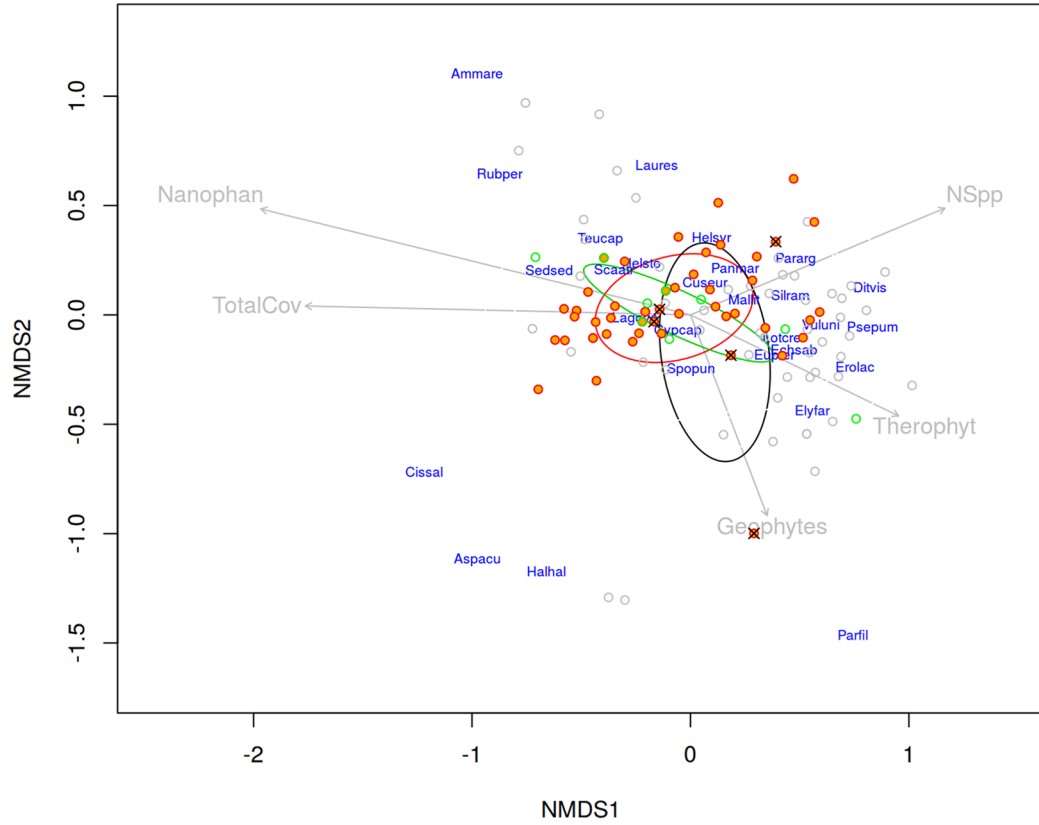

**Figure 6 Non-metric multidimensional scaling (NMDS) for accompanying species represented by pairwise Bray–Curtis distances between individuals (stress = 0.1413).** Orange filled points represent quadrats with the presence of diploid *C. aspera*, green with allotetraploid *C. seridis*, crosses with triploid hybrids, and empty circles quadrats with absence of *Centaurea*. Ellipses represent standard deviations for the three species. Arrows are fitted environmental variables. Analysis was performed in "El Saler North" site. Nanophan, nanophanerophytes; Theroph, therophytes; TotalCov, Total cover; Nspp, Number of species; Ammare, *Ammophila arenaria*; Rubper, *Rubia peregrina*; Laures, *Launaea resedifolia*; Teucap, *Teucrium capitatum*; Helsyr, *Helianthemum syriacum*; Sedsed, *Sedum sediforme*; Scaatr, *Scabiosa atropurpurea*; Panmar, *Pancratium maritimum*; Pararg, *Paronychia argentea*; Cuseur, *Cuscuta europaea*; Mallit, *Malcolmia littoralis*; Silram, *Silene ramosissima*; Ditvis, *Dittrichia viscosa*; Lagova, *Lagurus ovatus*; Cypcap, *Cyperus capitatus*; Vuluni, *Vulpia unilateralis*; Psepum, *Pseudorlaya pumila*; Lotcre, *Lotus creticus*; Echsab, *Echium sabulicola*; Spopun, *Sporobolus pungens*; Erolac, *Erodium laciniatum*; Elyfar, *Elymus farctus*; Cissal, *Cistus salvifolius*; Aspacu, *Asparagus acutifolius*; Halhal, *Halimium halimifolium*; Parfil, *Parapholis filiformis*.

diploid individuals were absent than in those where they were present. Furthermore, the quadrats with *C. aspera* individuals were significantly more distant to pathways than the quadrats without them. For *C. x subdecurrens*, only the percentage of hemicryptophytes slightly differed, but significantly, between the quadrats with the absence *vs* presence of individuals, although this result must be interpreted with caution because of few quadrats (4) there were where triploids were present. No significant differences for any environmental variable were found between quadrats with the absence *vs.* presence of *C. seridis*.

In relation to the accompanying species, *C. aspera* was positively/negatively associated with the species that were indicative of varied habitats and showed no particular ecological

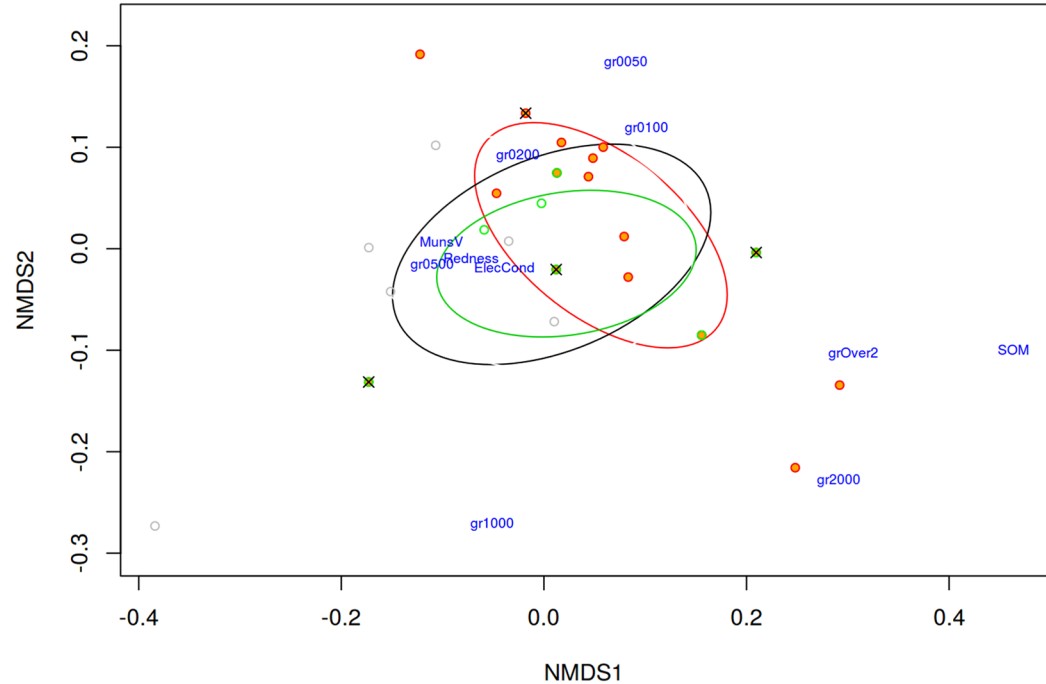

**Figure 7 Non-metric multidimensional scaling (NMDS) for soil variables represented by pairwise Bray–Curtis distances between quadrats (stress = 0.0562).** Orange filled points represent quadrats with the presence of diploid *C. aspera*, green with allotetraploid *C. seridis*, crosses with triploid hybrids, and empty circles quadrats with absence of *Centaurea*. Ellipses represent standard deviations for the three species. Analysis was performed in "El Saler North" site. gr2000, percentage of particles measuring between 1 and 2 mm in diameter; gr1000, percentage between 0.5 and 1 mm; gr0500, percentage less between 0.2 and 0.5 mm; gr0200, percentage between 0.1 and 0.2 mm; gr0100, percentage between 0.05 and 0.1 mm; gr0050, percentage <0.05 mm; MunsV, Munsell Value; SOM, soil organic matter; ElecCond, electrical conductivity.

pattern (Table 4). Specifically, it was associated positively with species with broad ecological requirements (*Lagurus ovatus* L., *Scabiosa atropurpurea* L.). It also correlated positively with *Helichrysum stoechas* DC. and negatively with *Erodium laciniatum* (Cav.) Willd., and both are indicative of semi-fixed dune habitats (*Costa & Mansanet, 1981*). This lack of ecological preferences by accompanying species was also evident in *C. seridis* and *C. x subdecurrens*, which showed no positive or negative correlations with any of the species present in the plot. Finally, no significant differences in relation to soil variables were found for any analysed taxon (Table 5).

## DISCUSSION

Although the diploid *C. aspera* and the tetraploid *C. seridis* displayed a wide overlapping distribution, they also showed relatively contrasting distribution patterns in east Spain on a broad scale. *C. seridis* displayed a narrower distribution area, confined mainly to the coast from Castellón to Gibraltar. Some individuals were also found inland, which agrees with previous works (*Merle, Garmendia & Ferriol, 2010*). Northwardly we did not find it, although it has been cited rarely in Catalonia (*Invernón & Devesa, 2013*). In contrast, *C. aspera* showed a broader distribution area, which covered the east half of Spain and

**Table 3 Multiple pairwise comparisons (Wilcoxon signed rank test with Bonferroni correction).**

| | C. aspera (2x) | | | C. seridis (4x) | | | C. x subdecurrens (3x) | | |
|---|---|---|---|---|---|---|---|---|---|
| | P-value (Wilcoxon test with Bonferroni correction) | Mean ± SE in quadrats with absence (60) | Mean ± SE in quadrats with presence (48) | P-value (Wilcoxon test with Bonferroni correction) | Mean ± SE in quadrats with absence (99) | Mean ± SE in quadrats with presence (9) | P-value (Wilcoxon test with Bonferroni correction) | Mean ± SE in quadrats with absence (103) | Mean ± SE in quadrats with presence (5) |
| Total vegetation cover | 1 | 49.3 ± 3.6 | 48.1 ± 3.6 | 1 | 48.37 ± 2.7 | 53.3 ± 8.0 | 1 | 48.2 ± 2.6 | 60.0 ± 13.4 |
| Nanophanerophyte cover | 1 | 11.6 ± 3.8 | 2 ± 1.9 | 1 | 8.0 ± 2.5 | 0 | 1 | 7.7 ± 2.4 | 0 |
| Hemicryptophyte cover | 1.3 e-08 | 5.9 ± 1.0 | 22.5 ± 3.2 | 1 | 12.8 ± 1.7 | 18.7 ± 10.0 | 0.017 | 11.5 ± 1.5 | 50.0 ± 14.9 |
| Chamaephyte cover | 1 | 22.7 ± 2.5 | 22.4 ± 2.6 | 1 | 22.8 ± 1.9 | 20.0 ± 5.3 | 1 | 23.1 ± 1.9 | 11.0 ± 3.6 |
| Geophyte cover | 1 | 7.0 ± 1.4 | 4.7 ± 0.9 | 1 | 5.3 ± 0.6 | 13.6 ± 7.5 | 1 | 5.7 ± 0.9 | 11.6 ± 5.3 |
| Therophyte cover | 1 | 4.3 ± 0.9 | 2.6 ± 0.3 | 1 | 3.6 ± 0.6 | 2.8 ± 1.1 | 1 | 3.6 ± 0.6 | 2.4 ± 0.8 |
| Path cover | 0.198 | 7.8 ± 3.0 | 0.1 ± 0.1 | 1 | 4.8 ± 1.8 | 0 | 1 | 4.6 ± 1.8 | 1.00 ± 0.0 |
| Distance to the nearest path | 3.36e-10 | 11.4 ± 2.2 | 46.2 ± 3.4 | 1 | 27.2 ± 2.7 | 23.1 ± 7.2 | 1 | 26.3 ± 2.6 | 39.4 ± 15.7 |
| Slope | 1 | 2.2 ± 0.3 | 2.2 ± 0.2 | 1 | 2.3 ± 0.2 | 1.6 ± 0.5 | 0.804 | 2.2 ± 0.2 | 3.2 ± 0.4 |
| Species richness | 0.0004 | 7.4 ± 0.3 | 9.4 ± 0.3 | 0.408 | 8.2 ± 0.2 | 9.8 ± 0.7 | 0.518 | 8.2 ± 0.2 | 10.4 ± 1.0 |

**Notes:**
Results of microspatial ecological parameters obtained in quadrats with and without diploids (*C. aspera*), with and without tetraploids (*C. seridis*), and with and without triploids (*C. x subdecurrens*) in "El Saler North."
SE, Standard error.

**Table 4 Chi-squared tests with Bonferroni correction.**

| Accompanying species | C. aspera (2x) | | | C. seridis (4x) | | | C. x subdecurrens (3x) | | |
|---|---|---|---|---|---|---|---|---|---|
| | P-value (Chi-squared test with Bonferroni correction) | Number of quadrats with accompanying species and absence of Centaurea | Number of quadrats with accompanying species and presence of Centaurea | P-value (Chi-squared test with Bonferroni correction) | Number of quadrats with accompanying species and absence of Centaurea | Number of quadrats with accompanying species and presence of Centaurea | P-value (Chi-squared test with Bonferroni correction) | Number of quadrats with accompanying species and absence of Centaurea | Number of quadrats with accompanying species and presence of Centaurea |
| Ammophila arenaria (L.) Link | 1 | 6 | 0 | 1 | 6 | 0 | 1 | 6 | 0 |
| Asparagus acutifolius L. | 1 | 1 | 2 | 1 | 3 | 0 | 1 | 3 | 0 |
| Cistus salviifolius L. | 1 | 0 | 1 | 1 | 1 | 0 | 1 | 1 | 0 |
| Cuscuta europaea L. | 1 | 2 | 0 | 1 | 2 | 0 | 1 | 2 | 0 |
| Cyperus capitatus Vand. | 0.223 | 36 | 41 | 1 | 70 | 7 | 1 | 72 | 5 |
| Dittrichia viscosa (L.) Greuter | 1 | 1 | 0 | 1 | 1 | 0 | 1 | 1 | 0 |
| Echium sabulicola Pomel | 1 | 4 | 5 | 1 | 8 | 1 | 1 | 9 | 0 |
| Elymus farctus (Viv.) Runemark ex Mederis | 1 | 13 | 5 | 1 | 16 | 2 | 1 | 17 | 1 |
| Erodium laciniatum (Cav.) Willd. | 0.006 | 35 | 10 | 1 | 43 | 2 | 1 | 44 | 1 |
| Euphorbia terracina L. | 1 | 23 | 19 | 1 | 36 | 6 | 1 | 40 | 2 |
| Halimium halimifolium (L.) Willk. | 1 | 3 | 1 | 1 | 4 | 0 | 1 | 4 | 0 |
| Helianthemum syriacum (Jacq.) Dum. Cours. | 1 | 7 | 9 | 1 | 15 | 1 | 1 | 16 | 0 |
| Helichrysum stoechas (L.) Moench. | 0.037 | 31 | 40 | 1 | 64 | 7 | 1 | 69 | 2 |
| Lagurus ovatus L. | 0.004 | 18 | 33 | 1 | 45 | 6 | 1 | 48 | 3 |
| Launaea resedifolia Druce | 1 | 2 | 3 | 1 | 5 | 0 | 1 | 5 | 0 |
| Lotus creticus L. | 1 | 44 | 31 | 1 | 70 | 5 | 1 | 70 | 5 |
| Malcolmia littorea (L.) R. Br. | 1 | 39 | 32 | 1 | 66 | 5 | 1 | 67 | 4 |

(Continued)

| Accompanying species | C. aspera (2x) | | | C. seridis (4x) | | | C. x subdecurrens (3x) | | |
|---|---|---|---|---|---|---|---|---|---|
| | P-value (Chi-squared test with Bonferroni correction) | Number of quadrats with accompanying species and absence of Centaurea | Number of quadrats with accompanying species and presence of Centaurea | P-value (Chi-squared test with Bonferroni correction) | Number of quadrats with accompanying species and absence of Centaurea | Number of quadrats with accompanying species and presence of Centaurea | P-value (Chi-squared test with Bonferroni correction) | Number of quadrats with accompanying species and absence of Centaurea | Number of quadrats with accompanying species and presence of Centaurea |
| Ononis natrix L. | 1 | 1 | 3 | 1 | 4 | 0 | 1 | 3 | 1 |
| Pancratium maritimum L. | 1 | 6 | 5 | 1 | 10 | 1 | 1 | 10 | 1 |
| Parapholis filiformis (Roth) C.E. Hubb. | 1 | 3 | 1 | 1 | 4 | 0 | 1 | 3 | 1 |
| Paronychia argentea Lam. | 1 | 12 | 17 | 1 | 26 | 3 | 1 | 28 | 1 |
| Phillyrea angustifolia L. | 1 | 2 | 2 | 1 | 4 | 0 | 1 | 4 | 0 |
| Pistacia lentiscus L. | 1 | 5 | 1 | 1 | 6 | 0 | 1 | 6 | 0 |
| Pseudorlaya pumila (L.) Grande | 1 | 8 | 1 | 1 | 8 | 1 | 1 | 9 | 0 |
| Rubia peregrina L. | 1 | 2 | 2 | 1 | 4 | 0 | 1 | 3 | 1 |
| Scabiosa atropurpurea L. | 2.0e-5 | 15 | 36 | 1 | 44 | 7 | 1 | 47 | 4 |
| Sedum sediforme (Jacq.) Pau | 1 | 18 | 24 | 1 | 36 | 6 | 1 | 40 | 2 |
| Silene ramosissima Desf. | 1 | 34 | 21 | 1 | 52 | 3 | 1 | 53 | 2 |
| Sporobolus pungens (Schreb.) Kunth | 1 | 32 | 36 | 1 | 61 | 7 | 1 | 65 | 3 |
| Teucrium capitatum L. | 1 | 9 | 1 | 1 | 9 | 1 | 1 | 9 | 1 |
| Vulpia unilateralis (L.) Stace | 1 | 26 | 14 | 1 | 35 | 5 | 1 | 38 | 2 |

**Note:**
Results of accompanying species observed in quadrats with and without diploids (C. aspera), with and without tetraploids (C. seridis), and with and without triploids (C. x subdecurrens) in "El Saler North."

**Table 5 Multiple pairwise comparisons (Wilcoxon signed rank test with Bonferroni correction).**

| | C. aspera (2x) | | | C. seridis (4x) | | | C. x subdecurrens (3x) | | |
|---|---|---|---|---|---|---|---|---|---|
| | P-value (Wilcoxon test with Bonferroni correction) | Mean ± SE in quadrats with absence (8) | Mean ± SE in quadrats with presence (16) | P-value (Wilcoxon test with Bonferroni correction) | Mean ± SE in quadrats with absence (17) | Mean ± SE in quadrats with presence (7) | P-value (Wilcoxon test with Bonferroni correction) | Mean ± SE in quadrats with absence (20) | Mean ± SE in quadrats with presence (4) |
| % Grains with diameter > 2 mm | 1 | 0.9 ± 0.4 | 2.8 ± 1.1 | 1 | 1.6 ± 0.6 | 3.5 ± 2.2 | 1 | 1.7 ± 0.5 | 4.4 ± 4.0 |
| % Grains with 1 < diameter < 2 mm | 0.556 | 0.2 ± 0.05 | 0.8 ± 0.3 | 1 | 0.7 ± 0.3 | 0.5 ± 0.2 | 1 | 0.6 ± 0.2 | 0.5 ± 0.2 |
| % Grains with 0.5 < diameter < 1 mm | 1 | 1.6 ± 0.6 | 2.0 ± 0.5 | 1 | 1.9 ± 0.5 | 1.7 ± 0.4 | 1 | 1.9 ± 0.5 | 1.9 ± 0.6 |
| % Grains with 0.2 < diameter < 0.5 mm | 0.144 | 76.8 ± 2.1 | 69.2 ± 1.5 | 1 | 70.7 ± 1.7 | 74.2 ± 2.6 | 1 | 70.9 ± 1.5 | 75.9 ± 4.2 |
| % Grains with 0.1 < diameter < 0.2 mm | 0.538 | 21.0 ± 2.5 | 27.0 ± 1.6 | 1 | 25.9 ± 1.7 | 22.8 ± 2.7 | 1 | 25.8 ± 1.5 | 21.1 ± 4.5 |
| % Grains with 0.05 < diameter < 0.1 mm | 0.054 | 0.4 ± 0.1 | 0.7 ± 0.1 | 1 | 0.6 ± 0.1 | 0.5 ± 0.1 | 1 | 0.6 ± 0.1 | 0.5 ± 0.1 |
| % Grains with diameter < 0.05 mm | 0.163 | 0.2 ± 0.05 | 0.3 ± 0.03 | 1 | 0.3 ± 0.04 | 0.2 ± 0.04 | 1 | 0.3 ± 0.03 | 0.3 ± 0.1 |
| Redness index | 0.628 | 3.9e-5 ± 5.2e-7 | 4.2e-5 ± 1.1e-6 | 1 | 4.1e-5 ± 1.2e-6 | 4.1e-5 ± 4.2e-7 | 1 | 4.1e-5 ± 9.8e-7 | 4.1e-5 ± 6.9e-7 |
| Munsell value | 1 | 6.0 ± 0.01 | 5.8 ± 0.1 | 1 | 5.8 ± 0.1 | 5.9 ± 0.1 | 1 | 5.8 ± 0.1 | 6.0 ± 0.0 |
| pH | 0.480 | 7.7 ± 0.1 | 7.4 ± 0.1 | 1 | 7.4 ± 0.1 | 7.6 ± 0.1 | 1 | 7.5 ± 0.1 | 7.6 ± 0.2 |
| Soil organic matter | 0.126 | 0.4 ± 0.1 | 1.8 ± 0.5 | 1 | 1.3 ± 0.5 | 1.2 ± 0.5 | 1 | 1.4 ± 0.5 | 0.7 ± 0.3 |
| Electrical conductivity | 1 | 0.1 ± 0.01 | 0.1 ± 0.01 | 1 | 0.1 ± 0.01 | 0.1 ± 0.01 | 1 | 0.1 ± 0.01 | 0.1 ± 0.01 |

**Notes:**
Results of microspatial soil parameters obtained in quadrats with and without diploids (*C. aspera*), with and without tetraploids (*C. seridis*), and with and without triploids (*C. x subdecurrens*) in "El Saler North."
SE, Standard error.

arrived at the coast, but only from Murcia northwardly (*Invernón & Devesa, 2013*). Thus on the coastline, a taxon distribution following a latitudinal gradient (diploid *C. aspera* to the north, tetraploid *C. seridis* to the south) was observed with a wide overlapping area. These results are similar to those observed in *Chamerion angustifolium* L. (*Sabara, Kron & Husband, 2013*) and *Actinidia chinensis* Planch. (*Liu et al., 2015*), in which the proportion of tetraploids in a population correlated negatively to latitude. Triploid hybrids arose whenever the distribution area of *C. seridis* and *C. aspera* overlapped, as previously observed in several contact zones near the coast (*Garmendia et al., 2010*).

On the coast, both the diploid *C. aspera* and the tetraploid *C. seridis* grew in nitrophilous and disturbed habitats due to grazing and human activities (tourism, urbanisation). This habitat was already described by *Rigual (1972)* 45 years ago, who found *C. seridis* plants growing on disturbed mobile dunes (*Sporobolo–Centaureetum seridis*, Rivas Goday & Rigual 1958) and in inland ruderal communities (*Asphodelo fistulosi–Hordeetum leporini* (*A. et O. Bolòs, 1950*) *O. Bolòs, 1956*). Although *C. aspera* has a wider ecological amplitude, it usually grows also in ruderal and nitrophilous inland and coastal habitats (*Invernón & Devesa, 2013*). In agreement with *Costa & Mansanet (1981)*, triploid hybrids *C. x subdecurrens* were found in the contact zones with heavy anthropogenic disturbance (*Centaureo maritimae–Echietum sabulicolae*, *Costa & Mansanet, 1981*), with high nitrification levels and several pathways used to reach the beach.

The polyploid complex composed of the diploid *C. aspera* and its derived allopolyploid *C. seridis* is another example of how disturbance can lead to the establishment of newly arisen polyploids (*Ramsey, 2011*; *Kim et al., 2012*; *Mráz et al., 2012*; *Soltis et al., 2015*). In the short term, the availability of new ecological niches may be a determining factor for the survival and long-term success of polyploids, which often occur more frequently in newly created, disrupted or harsh environments (*Van de Peer, Mizrachi & Marchal, 2017*). This is particularly true when polyploids are self-compatible as self-fertility promotes the colonisation of open patches (*Dorken & Pannell, 2007*). This is the case of *C. seridis*, which shows a high degree of autogamy, unlike *C. aspera,* which is obligately outcrossing (*Ferriol et al., 2015*). Otherwise in stable ecosystems, newly arisen polyploids may be unable to compete with their diploid relatives (*Van de Peer, Maere & Meyer, 2009*). Accordingly, tetraploids were found in higher proportions in the mixed-ploidy populations located in more disturbed habitats, which agrees with *Lumaret et al. (1987)* and *Mráz et al. (2012)*, who also found a higher proportion of tetraploids in more disturbed habitats due to human activities in *Dactylis glomerata* L. and *C. stoebe* L., respectively. Furthermore, the high frequency of *C. x subdecurrens* triploid hybrids can also be partly due to disturbance, which has been related with a higher frequency of triploids resulting from the hybridisation between diploid and tetraploid individuals (*Ståhlberg & Hedrén, 2009*).

The greater ability that polyploids display to colonise new habitats could be the result of adaptive processes, such as developing higher stress tolerance (*Van de Peer, Mizrachi & Marchal, 2017*). Consequently, diploid *C. aspera* and tetraploid *C. seridis* individuals may be differentiated according to habitat preferences, which allows their co-existence in
heterogeneous contact zones. In fact, this is one of the most cited mechanisms that facilitates the establishment and survival of neopolyploids in heterogeneous environments (*Ramsey & Ramsey, 2014*), such as dune fields, where local environmental factors, like soil and microclimatic characteristics, can vary on a scale of a few metres (*Linhart & Grant, 1996*). Along these lines, several examples that show a differentiation of related cytotypes on the microspatial scale in contact zones, according to different ecological factors, exist: microtopography and vegetation cover in *Senecio carniolicus* Willd. (*Hülber et al., 2009*); elevation and drainage patterns in *Taraxacum* sect. *Ruderalia* (*Meirmans et al., 2003*); level of shading in *Dactylorhiza maculata* (L.) Soó (*Ståhlberg & Hedrén, 2009*), or heterogeneity of habitats (presence of roads, forests, grasslands, and fields) in *Allium oleraceum* L. (*Šafářová & Duchoslav, 2010*). However in our case, neither *C. aspera* nor *C. seridis* was ecologically differentiated on the microspatial scale, which suggests lack of adaptive processes. Only for *C. aspera* were some significant differences found between quadrats with the presence and absence of individuals, but these differences were unrelated to any clear ecological pattern. Furthermore, the plants that were present or absent in the vicinity of *Centaurea* individuals did not show a clear ecological pattern altogether.

Despite there being no ecological differentiation between the diploid *C. aspera* and the tetraploid *C. seridis*, the individuals of the same taxon appeared to be significantly aggregated. A clumped distribution of individuals within a ploidy level seems a general rule in the studies of cytotype distribution on the microspatial scale, regardless of being ecologically differentiated or not (*Husband & Schemske, 2000*; *Johnson, Husband & Burton, 2003*; *Kolář et al., 2009*; *Trávníček et al., 2011*; *Laport & Ramsey, 2015*). In the *C. aspera*/*C. seridis* contact zones, the results suggest that the spatial aggregation of individuals of the same taxon has led to chance spatial associations with individuals of other species. This supports the existence of non-adaptive processes that result in the observed non-significant differences associated with species composition or ecological variables that characterise the niche of diploids and tetraploids, and also with a non-random *Centaurea* intraspecific distribution in contact zones. Firstly, the spatial aggregation of *Centaurea* individuals may be due to the low dispersion of achenes (*Li, Xu & Ridout, 2004*; *Baack, 2005*) or a short dispersal distance of pollen (*Fortuna et al., 2008*). Both *C. aspera* and *C. seridis* have persistent, short pappi that do not allow effective wind dispersal. In both species, the dry involucre retains fruits, so their dispersal depends on stem movements by either wind, or by passing animals, persons or vehicles (*Sheldon & Burrows, 1973*). Accordingly, they are considered to display atelechory, lack seed dispersal mechanisms and have short-distance seed dispersal (*García-Fayos, Engelbrecht & Bochet, 2013*), except ants, which may bring achenes into nests over longer distances (>1 m) (*Hensen, 2002*). Similarly, in spite of the lack of studies on *C. aspera* and *C. seridis* specifically, studies performed in other insect-pollinated *Centaurea* species have shown that most pollen grains disperse over short distances (<25 m), although a minor proportion can be dispersed further (*Hardy et al., 2004*; *Albrecht et al., 2009*). These short seed and pollen dispersal distances may, in turn, enhance intraspecific pollination and ultimately favour the co-existence of *C. aspera* and *C. seridis* in the absence of ecological

segregation (*Kennedy et al., 2006*). Secondly, tetraploid *C. seridis* individuals display high selfing levels (*Ferriol et al., 2015*), which can lead to the spatial segregation of taxa regardless of the niche differentiation among them, and can allow tetraploids to become established and survive (*Felber, 1991*). Thirdly, triploids are highly or completely sterile (*Ferriol et al., 2015*). Although varying degrees of fertility have been assessed in different triploid plant species, notably by autopolyploidisation (producing hexaploids) or by backcrossing with diploids (producing tetraploids) (*Ramsey & Schemske, 1998*), we did not find any hexaploid individuals among more than 220 individuals of the Moroccan and European populations (*Ferriol et al., 2012*; *Ferriol, Merle & Garmendia, 2014*). In forced crosses we observed complete sterility of pollen and ovules in the triploids from the "El Saler" population, also studied here (*Ferriol et al., 2015*). Thus the *C. x subdecurrens* individuals seemed to act as a strong triploid block. This strength of selection against triploids can also lead to clumped distributions by conferring spatial separation between parentals, and thereby reducing the competitive interactions between them and heteroploid crosses, which are the basis of the minority cytotype exclusion effect (*Hülber et al., 2015*). Other non-adaptive processes can promote the co-existence of the diploid *C. aspera* and the tetraploid *C. seridis* which cannot be ruled out are human-mediated colonisations by tetraploids. Similarly to that described for *C. stoebe* (*Mráz et al., 2012*), humans could have unintentionally dispersed tetraploid individuals into already established diploid populations by creating new open niches suitable for colonisation. Especially along paths and roads that run inwardly from the sea, the transport of tetraploid propagules like spiny capitula could have been facilitated by movement on pets and humans' belongings. Another explanation could be that plant populations have not struck the equilibrium at which all cytotypes but one are locally excluded (*Šafářová & Duchoslav, 2010*).

Even if the aggregated distribution of taxa may enhance the stability of ploidy co-existence by increasing the assortative mating rate in taxon-uniform clusters, hybridisation was not prevented. Triploid hybrids *C. x subdecurrens* are frequent in nature, and were found in all the mixed ploidy populations. On a fine scale, an intermediate spatial position between those of the diploid and tetraploid parentals should be expected, which agrees with *Ståhlberg & Hedrén (2009)*, who reported an intermediate position of triploid hybrids in mixed diploid/tetraploid populations of the *Dactylorhiza maculata* group, but with no statistical evaluation given the few triploids. However in our study, triploids appeared much closer to diploids than to tetraploids. This agrees with *Ferriol et al. (2015)*, who found that, due to the high degree of autogamy in the tetraploid *C. seridis*, and to the strict allogamy in the diploid *C. aspera*, triploid progeny always came from diploid maternal plants pollinated by tetraploid paternal plants in artificial crossings.

These asymmetric crosses, along with short achene distance dispersal and lack of ecological differentiation among taxa, could have led to a spatial distribution in which diploid *C. aspera* plants have to share space with their triploid offspring, while the *C. seridis* tetraploids can compete better for space. In addition to the high selfing rate, this better ability to compete for space compared with their diploid relatives could counteract the effects of the minority cytotype exclusion principle, and allow tetraploids to persist.

The mechanism by which diploids act as maternal plants and tetraploids as pollen donors by influencing the cytotype distribution pattern on the fine spatial scale, has also been suggested by *Suda et al. (2004)* in *Empetrum*. *Sabara, Kron & Husband (2013)* have also found that triploids are produced more often by diploid maternal plants than by tetraploids.

## CONCLUSIONS

In the *C. aspera* (2x)/*C. seridis* (4x) complex, adaptive mechanisms may exist that could lead to parapatric distributions on a broad scale to confine tetraploids to coast mobile dunes, while diploids develop inwardly from semi-fixed dunes. However, contact zones appeared, but only where dunes were strongly disturbed. Therefore, *C. aspera* and *C. seridis* coexist due mainly to non-adaptive mechanisms, and finally hybridise. In these contact zones, several mechanisms that allow the persistence of the tetraploid minor cytotype may take place. In addition to selfing and more assortative matings, the better ability to compete for space seems a key factor.

The results reported here can shed some light on the debate as to whether recently formed polyploid plants are evolutionary dead-ends (*Mayrose et al., 2011*; *Arrigo & Barker, 2012*) or, on the contrary, if they compete better than diploids (*Soltis et al., 2014*). Our observations support the idea that a large amount of neopolyploids, such as *C. seridis*, can overcome the minor cytotype exclusion, adapt quickly to new environments, and survive in the short term, although their long-term survival is still unclear (*Van de Peer, Mizrachi & Marchal, 2017*). Specifically, it has been shown in Asteraceae, which includes the genus *Centaurea*, that multiple WGD events have led to high rates of chromosome rearrangements and diversification, and finally to great evolutionary success (*Huang et al., 2016*).

### Funding
The authors received no funding for this work.

### Competing Interests
The authors declare that they have no competing interests.

### Author Contributions
- Alfonso Garmendia conceived and designed the experiments, analysed the data, contributed reagents/materials/analysis tools, prepared figures and/or tables, authored or reviewed drafts of the paper, approved the final draft.
- Hugo Merle conceived and designed the experiments, analysed the data, approved the final draft.
- Pablo Ruiz performed the experiments, approved the final draft.
- Maria Ferriol conceived and designed the experiments, analysed the data, contributed reagents/materials/analysis tools, prepared figures and/or tables, authored or reviewed drafts of the paper, approved the final draft.

## Data Availability

The raw data are provided in a Supplemental File.

## Supplemental Information

Supplemental information for this article can be found online at http://dx.doi.org/10.7717/peerj.5209#supplemental-information.

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
