# Peer review of "Distribution and ecological segregation on regional and microgeographic scales of the diploid Centaurea aspera L., the tetraploid C. seridis L., and their triploid hybrids (Compositae)"

_PeerJ, doi:10.7717/peerj.5209_

## Round 0.1 · original submission · Major Revisions

The reviewers were positive about your manuscript, and think it can be published, after making some improvements. Please address all the points raised by the reviewers point by point, referring to edits in the manuscript, or by explaining why you choose to not do so. After making your substantive edits, please try to get some help to strengthen the English writing.

Reviewer 1 ·

Basic reporting

In general, the manuscript reads well, it is well written, but it should be checked by a native English speaker because there are some hardly understandable sentences (lines 157-158, lines 267-268, among others).
The introduction is very well structured, but it would be necessary to add some reference on general polyploidy studies carried out by I.J. Leitch et al.
Authors should avoid redundancies between tables and text.

Experimental design

I think that the main weakness of this manuscript lies in the experimental design. After reading the previous papers of the authors and taking into account their expertise, I think that they can determine the cytotypes according to their morphological characters, but it would have been better to use the flow cytometry to determine the ploidy level of each individual included in this study.

Validity of the findings

Data have been well analyzed and the results correctly interpreted.

Additional comments

The authors should improve the conclusions section in order to avoid redundancies with the results section.
The manuscript is well structured and well written and it deserves to be published after minor revisions that are detailed in the preceding sections.

Reviewer 2 ·

Basic reporting

I have reviewed manuscript #26915 entitled ‘Cytotype distribution and ecological segregation on regional and microgeographic scales in Centaurea aspera L.-C. seridis L. (Compositae)’, co-authored by Garmendia and colleagues. The manuscript represents an interesting case-study aiming at getting further understanding of the potential role of ecological variables in shaping species distribution. I believe this manuscript is sound and fluently written, and as far as I understand, the statistical approach has been carefully designed.
However, my main criticism lays on the somewhat misleading understanding of the ‘cytotype’ concept in this manuscript. I am surprised that no methodological approach (e.g. cytotype screening by flow cytometry or chromosome counting) is taken to identify cytotype occurrence beyond a rather morphological description to allocate ploidies (if based on previous findings, then it should be referred accordingly with further details). In fact, on M&M, l153-onwards it is stated that cytotypes are attributed based on morphological traits solely. Bearing this in mind, one assumes that all C. aspera are (2n=20, 22), C. seridis (2n=44) and C. x subdecurrens (2n =?). If that’s the case, then I am wondering why using cytotype distribution in the title and not just ‘Distribution of C. aspera complex’? I would recommend the authors to clarify this point and address this issue on the introduction to avoid further misunderstandings along the manuscript. It seems that across the species complex there is some sort of chromosomal stability, but this point needs to be proven.
In addition to that, is it likely that tetraploid cytotypes of C. aspera could occur in the W Iberian Peninsula? The authors reported their presence on Morocco, and I see on the CCBD (http://ccdb.tau.ac.il/search/) that Humphries et al. also reported it previously. Could this eventual WGD occur elsewhere along the distribution area?
Concerning the triploid cytotypes (or C. x subdecurrens). So far I believe this cytotype has been only recorded by means of DNA ploidy allocation, but no chromosome work has been reported so far to either (i) understand its meiotic behaviour, or (ii) to confirm the chromosome number and get insights into the cytogenetics complement of the parental donors (e.g. 2n=20 + 44/2n= 22 +44). Is there any chance that 3x individuals could overcome sterility through production of unreduced gametes and therefore hexaploid cytotypes might exist, even if they could be extremely rare? Even if this has not found yet, one would appreciate some discussion on the text. Spontaneous and induced (auto)polyploidization is something reported in plants as a mechanism to overcome sterility in triploid cytotypes. Obviously, natural recovery of fertility is rare, but worth exploring in detail on those 3x-ploidy allocated populations. I understand this might fall outside the main scope of this work, but a detailed cytogenetic exploration of 3x-populations could shed light into the ongoing mechanisms shaping (or preventing) its current distribution.
In my opinion, Figures 7 and 8 would benefit from adding further details on the caption. As they currently state, I find them difficult to interpret (e.g. I assume that filled/empty/crossed data points are cytotyes, but cannot differentiate among them beyond the triploid (assumed x)).

Experimental design

See my basic reporting section. I believe the analytical approach is sound and robust.

Validity of the findings

See my comments on the basic reporting section

Additional comments

See my comments on the basic reporting section

·

Basic reporting

Word choice and syntax throughout this manuscript are often unusual and occasionally make the work hard to follow. I would suggest the authors reread this with an eye for clarity and potentially enlist someone for whom English is their first language. I have made specific comments on particular phrases in the line edits, but these should not be considered comprehensive. A general revision of the English would greatly improve the manuscript’s clarity and ease of reading.

In general, the literature is sufficiently cited, though I made comments to authors where some additional reading may be necessary.

The figures which are present are neat and clear, but some figures and tables are either unnecessary or could be pared down from their current sizes.

The manuscript is self-contained and internally consistent, though the distributions of cytotypes reported here aren't that novel given previous work by these authors.

Experimental design

The primary concern I have with this study is the treatment of density of each cytotype relative to the others. The method used here, coarse binning into three categories followed by chi-squared tests, is not described in sufficient detail for its biological relevance to be determined. There are no citations given in the methods or discussion linking this method to anything in previous literature, leaving me with the impression that it is of the authors’ own design. If that is not the case, it needs justification based on the literature. If it is the case that this is a novel way to describe relative density of pairs of cytotypes, it is on the authors to establish its biological relevance. Given that the three plots used for microspatial analysis differ in density by an order to magnitude in some cases, the authors do not successfully make the case that equal-area bins of relative density as described here are relevant to competition between cytotypes. Either further literature review needs to be conducted and cited to justify their approach, or they need a new approach. It seems that an adaptation of neighborhood analysis (along the lines of Silander and Pacala, 1985 or Sanderson et al., 2016, Ecology and Evolution—subbing in cytotype in place of sex) might allow for finer description of relative densities of each cytotype and allow for direct comparison between populations. To my knowledge, such an application to a polyploid complex would be quite novel if the authors chose to pursue it, and it looks like they have the data to do so. Related but a bit tangential, I don’t find Figure 5 particularly informative, as it basically recapitulates what is shown in the fine-scale mapping in Figure 2.

While the treatment of local density is my primary concern, I also wonder why the authors focus on the relatively broad life-form categories when they have species-level data that actually shows some significant differences. Apologies if this is a question from ignorance, but could the NMDS analyses be re-done using these species-level associations? It is possible that the authors have done so but not reported it here, but if not, perhaps worth considering?

Lastly for major concerns of the paper, I feel that the interpretations of the close-proximity of diploid and triploid individuals is not fully realized in light of what is known about polyploid complexes and interploid reproduction. The association of triploids with diploid moms would be highly unusual if one did not know that tetraploids are largely autogamous. Typically crosses between diploids and tetraploids fare better when the maternal plant is tetraploid, and it might be worth exploring why the opposite is true here. Not merely due to tetraploid autogamy: the abundance of triploids suggests that these crosses are relatively successful, when they are often not in other systems! Also, while the authors go into detail about seed dispersal distance and how short dispersal distances may facilitate clumping and therefore foster cytotype coexist, they say very little about pollination or pollinator behavior. If diploids are at risk of being swamped out by tetraploid or sterile triploids, distances over which pollinators travel can matter greatly, and that isn’t addressed or even referenced in citation here.

Validity of the findings

Given that novelty is not assessed and negative or inconclusive results are allowed, I have no comments on that aspect of the work.

The data presented are robust and statistically sound.

The conclusion is well-stated, though it could be improved with stronger connections to the existing interploid reproduction literature.

Additional comments

All line comments are as follows. While these are largely grammatical, some are non-trivial.

Line comments:

Line 43-45: There are a number of recent phylogenomic papers that could (and I think should) be included in a discussion of the overall success or failure of polyploidy. Mayrose et al., 2011 and Soltis et al., 2014 would be good places to start.

Line 46: Arrigo and Barker is not the best citation for this idea and is only a secondary concept of that paper. The Levin or Petit, Bretagnolle, and Felber papers cited therein would be better, as would some of Husband’s work in Chamerion angustifolium.

Line 53: confer onto them, not confer them

Line 56-57: Zozomová-Lihová is the correct citation, yet you make this two people five words later. In general, citations throughout the manuscript are rather sloppy.

Line 63-64:
“even in the separation of cytotypes into formations of a different physiognomy” This phrase is lifted almost word-for-word from Hülber et al., 2015. Revise.

Line 67: vegetative reproduction isn’t adaptive?? Either cite something that specifically goes to this point or revise.

Line 72-73: Herben, Trávníček, and Chrtek (2016) might be worth reading and citing here.

Line 75: fewer, not less

Line 79-80: “polyploids used to be separated from their diploid ancestors” I’m not sure what you mean here, and this might just be a wording issue. Neopolyploids almost invariably start out sympatric to their diploid progenitors given how they usually form. If you feel this phrase is accurate, its meaning is unclear to the reader.

Line 84: No space in McAllister

Line 89-90: Odd phrasing: “can allow to infer”. Try “can allow one to infer” or “can allow inference of”

Line 108-109: That’s quite a few outlying sites if you want to make the case that C. seridis has a more limited distribution. Perhaps give a sense from these studies how common it is in these locations. Are we talking single populations? Also, do you mean Albania, not Albany?

Line 110: Generally the –ern ending is used on directional adjectives (i.e. eastern, western). Using east and west as done here is understandable and should be fine, but be consistent throughout the manuscript.

Line 122-124: I’m a bit dubious on this as an example of assortative mating that would preserve cytotypes. Yes, the tetraploids are expending fewer resources than the diploid maternal plants are, but pollen expenditure is still a cost.

Line 147: “Here the Mediterranean coast is particularly stressed as it is the typical habitat of C.
seridis” This phrasing is unclear. What do you mean by stressed? And it sounds as if it is stressed because it is the habitat of C. seridis. Consider revising.

Line 151-153: Do the salt-marshes prevent formation of these dune structures or simply fail to occur where dunes are present? The phrase “avoid the presence of” makes this unclear.

Line 171: For North American publications, 2.400 should be 2,400.

Line 175: Orientation is somewhat unclear to me. By orientation, do you mean slope aspect, or something else?

Line 177: This may be overly pedantic, but how does one take the central quadrat from a 100-m2 area (or what I assume is a 10-quadrat x 10-quadrat area?). It sounded like, based on the methods thus far, that the quadrats were already laid out, so did this “quadrat” span four existing ones at the center of each area? Again, this may be overthinking.

Line 193: Did you mean sand here, and not soil? If so, how was this retain (I assume following the final sieving?). This might be worth briefly pointing out for the uninitiated.

Line 199-217: I’m not sure listing the software used to prepare the document is necessary, and writing out all software used at the beginning of the section makes it hard for the reader to follow. I recommend cutting all or most of this paragraph and citing each software package as it comes up in the text, which you’ve already done to a large extent.

Lines 249-263: Unless this is the first reporting of some of the details (overall distribution of each species, which is a dune specialist, etc), these details belong in methods under a “study system” heading or something similar.

Line 349: conclusive, not concluding

Line 376: find, not found

Line 395, Figure 2: It would really be helpful if, on the fine-scale maps, the authors could indicate the locations of pathways used to access the beach. It would make it easier to fully grasp their effect on C. x. subdecurrens and C. seridis.

Line 436: Both Kolář and Trávníček need to be corrected.

Line 448: for should be over

Line 465: “could have been facilitated” does not work here in the sentence. Perhaps, “the transport of tetraploid propagules could have been facilitated by movement on pets and human belongings”

Line 469: two “nots” in a row

Tables 3, 4, 5: I’m not sure why we’re being shown uncorrected P-values. Just show us the Bonferroni-corrected values.

---

## Round 0.2 · accepted · Accept

Thank you for your careful and elaborate responses to the reviewers' comments.

#